# Healing Bodies, Healing Communities: A Community-Based Qualitative Study of Adult Survivors of Childhood Sexual Trauma in South Africa

**DOI:** 10.3390/healthcare13202601

**Published:** 2025-10-15

**Authors:** Leona Morgan, Sarojini Nadar, Ines Keygnaert

**Affiliations:** 1Faculty of Arts and Humanities, University of the Western Cape, Cape Town 7535, South Africa; snadar@uwc.ac.za; 2International Centre for Reproductive Health (ICRH) and VIORESC, Faculty of Medicine and Health Sciences, Ghent University, 9000 Ghent, Belgium; ines.keygnaert@ugent.be; 3Department of Obstetrics & Gynaecology, Ghent University Hospital, 9000 Ghent, Belgium

**Keywords:** adult survivors, childhood sexual trauma, integrated trauma-informed care, critical feminist praxis, therapeutic care pathway development, marginalized contexts, systemic violence

## Abstract

**Highlights:**

Body-based care models respond better to long-term, intergenerational and somatic aspects of sexual trauma in survivors being historically excluded from mental health care. Co-creation of care pathways ensures culturally sensitive approaches that are responsive to lived experiences of marginalized survivors of childhood sexual trauma.

**What are the main findings?**
Relational safety and somatic engagement were foundational to trauma recovery, enabling survivors to process trauma at their own pace and integrate dissociative experiences through embodied therapeutic praxis.Recovery was relational and continuous, with participants reporting increased peace, authenticity and social connection despite structural barriers, highlighting the effectiveness of culturally grounded, non-pathologizing care.

**What is the implication of the main finding?**
Integrative Trauma-Informed Care (ITIC) offers a culturally sensitive, adaptable framework that can be tailored to diverse communities and age groups, emphasizing embodied, intergenerational and relational healing.Decolonial and feminist approaches to mental health care can bridge epistemic gaps in ITIC praxes by centering survivors’ lived, embodied experiences, promoting sustainable and inclusive therapeutic models globally.

**Abstract:**

Background: While sexual trauma is inherently an embodied experience, research on psychological interventions that is cognisant of geographic and socio-political community contexts within which embodied, therapeutic interventions occur remains limited. Decolonial, African and feminist community psychologies have noted this epistemic–ethical gap. Objectives: This paper explores the co-development of trauma-informed care pathways for adult survivors of childhood sexual trauma (CST) in under-resourced communities in Cape Town, South Africa. The study aimed to integrate intergenerational community knowledge, embodied therapeutic practices and collaborative approaches into locally relevant models of care. Methods: Drawing on feminist mental health frameworks, this qualitative study engaged 13 adult female survivors who identify as “coloured”. Embodiment was central in guiding the deconstructive therapeutic praxis, informing both the co-development of care pathways and the conceptualization of integrative trauma-informed care (ITIC) beyond pathologizing, deficit-based narratives. The cultivation of trust and the situated lived realities of survivors were foregrounded to illustrate the relational dimensions of trauma recovery. Results: Establishing relational safety emerged as the foundation for therapeutic engagement, supported by non-directive therapeutic probing. Grounding practices, affective regulation and embodied awareness enabled participants to process trauma at their own pace. Somatic engagement allowed the integration of dissociative experiences while strengthening relational resilience. Recovery was a continuous process, with participants reporting increased peace, authenticity and capacity for social connection despite structural barriers to community support. Conclusions: The development of care pathways was embedded within the research process itself, offering an approach that is culturally sensitive and responsive to survivors’ lived experiences. ITIC accounted for temporal, intergenerational and embodied trauma and should be adaptable across age and community-specific needs. The ITIC approach offers a transferable framework for co-developing de-pathologizing, culturally responsive interventions that can be adapted across diverse global contexts to support sustainable trauma integration.

## 1. Introduction

The long-term physical and mental health consequences of traumatic experiences, particularly those occurring in childhood, are well documented [1,2,3]. Among these, childhood sexual violence stands out as especially harmful, with lasting impacts that extend well into adulthood [4,5,6]. These effects are often exacerbated in marginalized contexts, where structural inequalities compound physical vulnerability and increase the risk of exposure to multiple forms of violence, including childhood sexual trauma (CST) [7]. Accordingly, the development of trauma interventions that are grounded in the contextual realities of victim-survivors from such communities is essential. This need is especially pressing considering the growing shift in psychosocial trauma-informed care toward integrative and contextually grounded mental health care practices that acknowledge diverse epistemologies in health and social sciences [8,9].

In South Africa, health psychology emerged during the transition from apartheid’s entrenched structural inequalities after 1994, in response to urgent public health needs [10]. This historical context necessitates transformative shifts in health care as persistent governance challenges rooted in apartheid-era nepotism and exacerbated by post-apartheid corruption and neoliberal policies continue to hamper access to specialized trauma care in many communities [11,12]. The First South African National Gender-Based Violence Study [13] highlights the alarming prevalence of sexual and physical violence against women. Based on nationally representative data, 35.5% of women reported experiencing physical and/or sexual violence in their lifetime, affecting an estimated 7.85 million women. Experiences of violence varied significantly by age, race, employment status and current relationship status. Notably, 58% of women reported physical abuse before the age of 15—equivalent to approximately 12 million individuals—while 4% reported childhood sexual abuse (CSA) before the age of 15, affecting an estimated 880,530 women. These findings indicate the widespread scale and early onset of sexual violence experienced by South African children and women and the critical importance of developing responsive, community-centered trauma care. Scholars in critical feminist psychologies have foregrounded the need for more equitable healthcare [14,15] as these enduring inequalities, intersecting with the historical marginalization of women and girls, have contributed to the ongoing prevalence of sexual violence in the country. In addition, scholars [16,17,18] highlight psychologies that prioritize relational healing, contextual safety and the co-construction of meaning, while actively resisting pathologization in promoting culturally grounded therapeutic practices.

As a result, this qualitative, community-based study is grounded in the intergenerational lived experiences of adult victim-survivors of CST on the Cape Flats, a region that has been shaped by systemic marginalization, historical displacement and ongoing structural violence. The Cape Flats is situated to the south-east of Cape Town’s central business district and is the result of race-based legislation that enforced a divided society [19,20,21]. Thriving communities were forcefully relocated out of more central urban areas designated for white people, with no warning and often overnight. These relocations often divided members of the same families by racial categories and skin tone, into government-built townships on the Cape Flats. Described as “the dumping grounds of apartheid”, the phrase is often attributed to various scholars, activists and writers who have described the Cape Flats in the context of its forced segregation during apartheid. Social and economic marginalization persist for many residents, while they also deal with high incidences of crime and gang-related violence [7,22]. Scholars have noted the effect of community violence and continued systemic neglect and poverty, with a specific impact on the well-being and quality of life for the unemployed youth population [23]. Community-specific considerations, described as “an ecosystem of violence” ([24], (p. 2)), contributed to the lived experiences of sexual violence for the 13 women who participated in this study.

Considering these contextual and theoretical considerations, this paper explores the co-development of contextually grounded, embodied trauma care practices that are culturally relevant and ethically collaborative in communities historically denied access to effective mental health care services. As trauma integration is a complex and multifaceted process, integrating personal narratives while assessing the physiological, emotional and cognitive impacts of trauma suppression is essential [25,26,27]. In doing so, the study is committed not only to generating knowledge but also to advocating for integrative trauma-informed care (ITIC). Trauma-informed care (TIC) emerged in the early 2000s [28]. Building on this foundation, recent scholarship has expanded TIC into more holistic and integrative contextual-sensitive models. ITIC frameworks extend beyond individual treatment to address trauma across ecological levels, including collective, systemic, historical and intergenerational dimensions. Notably, the Integrative Trauma and Healing Framework [29] defines trauma as a disruption of safety, agency, dignity and belonging, and conceptualizes healing as the restoration of these core needs. TIC is increasingly seen as essential in primary care, though gaps in implementation and training remain significant. We argue that ITIC must be adapted to the specific needs of sexual trauma victim-survivors to be effective and just, while being constantly attentive to researcher reflexivity. There is a need for care models that resist universalized and standardized framings of CST and instead open space for alternatives that are contextually grounded, co-constructed between survivors and therapists and responsive to survivors’ own knowledges; innovative qualitative methodologies are thus critical for tracing how personal experiences of harm are entangled with wider structural conditions of poverty, gendered violence and social exclusion, and for imagining forms of care that are both just and transformative [15]. We suggest that both the findings and the framing are applicable for global contexts. A decolonial feminist framing might appear at first glance to be tied only to the Global South, but its value extends far beyond that geography. Decoloniality is not solely about national liberation from colonial rule; it is a critical method for interrogating the structures of knowledge, power and subjectivity that continue to shape people’s lives worldwide. For adult survivors of CST, these structures matter because trauma is never experienced in isolation. It is mediated through cultural narratives about gender, shame, purity, family and belonging—all of which are historically entangled with colonial and patriarchal systems of control. Even in contexts not marked by formal colonial histories, survivors encounter forms of epistemic violence: silencing, disbelief, pathologization or the reduction of their experiences to clinical categories that erase embodied and cultural, relational dimensions of harm. A decolonial feminist framing resists this flattening by foregrounding survivors’ voices, contextual knowledges and lived realities as legitimate sources of meaning.

The objectives of evaluating the co-development of the therapeutic care pathways were three-fold: first, to foreground victim-survivor knowledge and the cultural context; second, to prioritize relational trust, particularly in communities historically excluded from mental health systems; and third, to investigate the benefit of integrative, flexible body-centered care models that are responsive to long-term, intergenerational and somatic aspects of trauma. Highlighting the importance of actively involving survivors in this collaborative process, this study’s aim was specifically to describe the importance of co-creating adapted and context-sensitive therapeutic care pathways as part of ITIC models.

## 2. Materials and Methods

This study was shaped in three ways to include reflections on the practitioner–community relationship, firstly, by actively adopting an epistemological framing that was located within feminist critical psychology. Secondly, this framing opened space to adopt a critical reflexivity for the researcher, that actively gave voice to her socio-political location and community-centered epistemic positioning. Thirdly, “embodiment” was centralized in the feminist therapeutic encounter, not as diagnostic but as deconstructive in the way the therapeutic care pathways were co-developed and ITIC conceptualized outside of restrictive or pathologizing narratives [30].

### 2.1. Study Design

This exploratory qualitative study used a community-based, small-scale, cross-sectional design with non-probability purposive sampling to engage a hard-to-reach population of adult female survivors sharing a specific gender identity and socio-cultural background. Therapeutic encounters functioned as both care and data collection points, enabling a participatory and clinically informed exploration of lived experiences of CST, grounded in critical psychology and decolonial feminist epistemologies. The methodology foregrounds a co-creative process of knowledge production through relational engagement and situated reflexivity, enabling the emergence of embodied ITIC pathways shaped by each participant’s lived realities. This approach offers a contextually grounded alternative to disembodied models of trauma care. Applied with critical attentiveness, it highlights best practices for building transparent, authentic and ethically sound therapeutic relationships with adult victim-survivors of CST affected by the intergenerational impact of structural violence and spatial precarity on the Cape Flats. From a critical decolonial feminist psychology perspective, attending to power dynamics and researcher positionality are essential to developing equitable and collaborative ITIC pathways [31,32].

### 2.2. Facilitating Safe Participant Inclusion

Participant inclusion followed a relational, community-engaged approach, consistent with a co-creative and ethically accountable methodology [33,34]. Potential participants were primarily referred through churches and interfaith activist networks on the Cape Flats, following presentations by the first author at community events in Bonteheuwel and Heideveld. These gatherings facilitated transparent discussions about this study’s aims, with particular attention given to what co-development of therapeutic care pathways entails. Central to these conversations was the exploration of the physical and mental health dimensions of long-term sexual trauma recovery with an emphasis on the meaning of embodied and reflexive therapeutic engagement. Participation was voluntary, and 13 women who had experienced CST across different developmental stages of their lives chose to take part. To uphold principles of social justice and equitable access, all therapeutic sessions were provided free of charge, ensuring the study’s commitment to sustained trauma support and ethical research practice. The first author conducted 73 sessions in total from March to November 2022. A detailed account of participants’ micro-, meso-, exo- and macro socio-ecological contexts has been comprehensively outlined by Morgan et al. ([34] (pp. 4, 7)).

### 2.3. Researcher Positionality and Reflexivity

The primary researcher’s clinical experience was guided by the ethical code of professional conduct guidelines outlined by the Health Professions Council of South Africa [35]. The primary researcher, a South African clinical psychologist, conducted this study as part of a joint PhD in ethics (UWC) and health sciences (UGent). Reflexive praxis was critically engaged with throughout this study. This involved actively working towards decolonizing the research process and addressing the power imbalances that persist in certain post-apartheid contexts. Scholars in feminist decolonial studies argue that the politics of knowledge production aim to include researcher reflexivity as critical—not only in acknowledging positionality but also in navigating the entanglements of power, ethics and representation in research [36]. This framing demands a conscious effort to move beyond disciplinary orthodoxies by centring embodied methodologies aligning with the lived complexities of oppression, resistance and trauma recovery. As both method and praxis, community-centred ITIC approaches invite participants to construct meaning from their lived experiences and narratives, guiding the embodied trauma recovery process. ITIC thus becomes a transformative act, making visible the intersections of historical trauma, structural violence and everyday life. By embedding researcher reflexivity and embracing collaborative modes of inquiry [37], the first author engaged in two main forms of reflexive practice. First, through introspective, embodied and intersubjective reflexivity, she remained attentive to how her presence as a white, Afrikaans-speaking therapist was experienced by participants, drawing on ongoing reflections and dialogical engagement with them. As a cultural outsider, this positionality carried racial and socio-economic privilege, which required careful negotiation of power dynamics and trust. Her clinical expertise, feminist–decolonial orientation and professional capacity and experience to create safe spaces for ITIC were strengths that supported both the research and therapeutic processes. Engagement with complex trauma and structural inequalities carried significant emotional weight, making therapist self-care essential. Supportive supervision, collegial dialogue and embodied practices of grounding and self-reflection were employed to manage vicarious distress and sustain relational presence. This reflexive and self-care stance was integral to ensuring ethical, sustainable and participant-centred modes of knowledge production.

### 2.4. Embodiment: Central to Co-Creating Data

Framed within a critical feminist praxis, this study was designed as a collaborative process in which participants actively co-constructed knowledge with the therapist-researcher. Specifically, the study employed iterative, dialogical interviews in which survivors were invited not only to share their narratives but also to reflect on the themes as they emerged. Participants also contributed to discussions about the implications of the findings for care which informed the development of alternative pathways of support that foregrounded their lived realities and priorities. This participatory approach ensured that the resulting models of care are not only contextually grounded but also shaped by those whose lives they are intended to serve. The data collection process allowed a distinct, unique and independent coming to voice and expression of lived experiences of severe and repeated sexual victimization for each participant. This perspective acknowledges the shaping of both the content and methodology of data production, by generating data using several different methods, starting with an open-ended clinical interview. The interview process was designed to account for the effects of long-term trauma suppression, non-disclosure and learned helplessness, prioritizing the timing of questions and non-verbal affirmations, as summarized in Table 1. Applying clinical expertise was critical for complex trauma-informed assessment and the therapist-researcher was cognizant of the unique embodied trauma integrative process of each participant [38], as summarized in Table 2.

The second method involved reflective questioning and detailed process notes, emphasizing rapport-building in an open-ended manner. Pathologizing language such as “toxicity” and “triggers”, which could reinforce victimhood and detract from the participants’ lived experiences and agency, was consciously avoided. Instead, affirmations of participants’ lived experiences were prioritized through reflective affirmations. A third method, which constituted the bulk of the data production, involved follow-up sessions that explored trauma integration themes and intervention timelines in detail. To create a safe and supportive therapeutic space within the church office and madrassa where the ITIC sessions were facilitated, some sessions were conducted while participants lay on a comfortable couch or bed, with cushions and blankets provided as needed.

Taking these methodological considerations into account for data analysis, audio-visual recordings were made of the facilitating sessions, with adherence to ethical guidelines for encryption, password protection and ensuring the confidentiality of raw data, thereby respecting the informed process of consent provided by the study participants.

### 2.5. Data Analysis and Coding 

Qualitative, in-depth inductive analyses were conducted, including semantic and latent content. The analysis process used inductive methods by generating insights directly from data, rather than imposing pre-existing constructs or theories. Applying the six phases of reflexive thematic analysis [39], nuanced themes, concepts and categories in a broad overview of each participant’s process were identified and coded. This included integrating clinical observations (body posture, tone of voice, silences and physiological cues) to capture the embodied dimensions of trauma disclosure and integration. The analysis proceeded in three main stages. First, through data immersion the first author engaged repeatedly with transcripts, audiovisual recordings and clinical notes, manually coding for verbal and non-verbal indicators of trauma processing. Reflective memos documented analytic decisions and emergent insights. Second, theme development and validation were reached as codes were iteratively clustered into themes, integrating narrative accounts, somatic descriptions and relational dynamics observed during sessions. This phase included cross-checking patterns against clinical observations (e.g., shifts in body posture or breathing) to ensure analytic depth. Themes and constructs were identified using a bottom-up approach, keeping the themes close to the interviews and ITIC sessions. Coding validity was established by cross-checking the initial coding process and applied conceptualizations of constructs. Main themes, concepts, non-verbal cues and silences were coded for according to session numbers and the total intervention timelines. Once a theme was identified, a time sequence of integration was identified that varied for each participant. Timestamps in the total intervention timeline were coded for to highlight the importance of time- and cost-effective trauma-informed care, which is detailed in Appendix A.

Themes were collaboratively refined through peer debriefing with co-authors to enhance interpretive rigour and to verify coherence with the lived realities and individual participant qualitative and in-depth care pathway co-development processes described in Appendix A. A phased summary of the main themes identified in the therapeutic timeline is described in Figure 1. Qualitative analytical rigour was supported through triangulation of data sources (transcripts, session recordings, clinical notes) and iterative reflexive memo writing. Transferability was strengthened by providing rich, contextualized descriptions of participants’ socio-cultural and clinical contexts. Dependability and confirmability were ensured by maintaining a transparent audit trail of coding decisions, analytic memos and research team discussions.

The authors identified five main themes through discussions of the identified codes and themes, consensus-building and an iterative process, forming part of the ITIC process analysis as shown in Figure 2.

## 3. Results

In total, 13 in-depth initial interviews and 60 follow-up sessions were recorded, with the number of follow-ups tailored to individual participants’ availability. The 73 sessions totaled 53.3 h and 885 transcribed single-spaced pages. The initial analysis and coding were grouped into preliminary themes as illustrated in Figure 2, which were critically reviewed, refined and interpreted in relation to the overarching theoretical frameworks of decolonial feminist psychology and ITIC. The colour-coded themes and subthemes summarizing the unique, non-linear therapeutic care pathway co-development process of each participant (who are referred to by using pseudonyms), are summarized in Appendix A. The content of Appendix A is categorized to reflect the multifaceted nature of the therapeutic process, offering a structured overview of its stages, planning steps and main objectives. Each row and column corresponds to a specific methodological phase in column 2, which is applied in facilitating and analyzing the integrative use of silence in context-specific, complex trauma recovery with the same cohort of participants by mapping trauma integration across eight methodological non-linear phases ([34] (p. 5)). Appendix A highlights moments when significant therapeutic themes emerged, affective shifts occurred and the therapist responses facilitated and supported trauma integration. Timeline and phase indicators trace each participant’s progression through trauma disclosure, emotional processing, memory recall and the integration of embodied awareness. Appendix A also illustrates how trauma-related themes such as fear, anger, sadness, isolation and empowerment developed across sessions, supporting participants’ emotional processing through embodied, phase-based care. Each pathway is described in terms of affective valence—the emotional tone or intensity underpinning the therapeutic engagement, highlighting how participants’ expressions of distress, resilience or ambivalence shaped the therapeutic context. The analysis attends to shifts in participant self-awareness, exploring how personal trauma narratives evolved as the study participants reflected on their histories of CST within a safe and responsive therapeutic setting. This include physical health indicators such as chronic pain, fatigue and sleep disturbances that were frequently exacerbated by the long-term impact of adverse childhood experiences (ACEs) and age-related vulnerability.

### 3.1. Establishing Relational Safety

A central theme that emerged from the analysis of ITIC sessions was the establishment of relational safety, which served as the foundational condition for participants’ engagement in trauma integration processes. Three main subthemes were identified under this broader theme, namely, trauma-informed consent, which foregrounded participant autonomy and collaborative decision-making; spontaneous trauma disclosure, which often occurred organically once a sense of trust was established; and non-directive therapeutic probing, which allowed for deeper exploration without imposing interpretation or re-traumatization. These elements collectively supported the development of a relationally attuned, participant-led recovery process.

During the first semi-structured interview, written informed process consent was obtained to ensure that participation was voluntary and not influenced by the referral process. The first author approached informed consent as an ongoing, collaborative process of mutual agreement, rather than a one-time act of signing a consent form at the start of the study. Forms specifying the quantity and duration of sessions and interviews were signed after each session and interview to ensure transparency. The semi-structured format of the first interview ensured the establishing of rapport and therapeutic relational trust as the severity of the violence experienced by the research participants necessitated this nuanced, clinical trauma assessment process. Emphasis was placed on creating a safe and non-judgemental space, supported by this flexible assessment process. Clinical expertise in applying this flexibility was essential for determining the appropriate timing and structure of questions related to participants’ clinical histories and lived experiences of sexual violence. The assessment included comprehensive information about demographics, family history, developmental trajectory from childhood through adulthood and their current psychosocial functioning, encompassing physical and mental health, daily activities and trauma-related challenges. Detailed clinical intake forms, which served as structured clinical assessment tools in this study ([34], Annex 1), may be considered as reference in similar trauma-informed intervention contexts. Specific attention was given to participants’ consciousness of their childhood trauma at the time of the interventions process. This is exemplified by the reflection of Evelyn, who experienced repeated sexual abuse by a friend of her father’s, at the age of eight years:


*“No, it never goes away … the memories.”*


Ava, who had been abused from the age of eight years by her brother and father and gangraped in a public space at the age of 40, stated:


*“I was raped by my own brother when I was eight; I remember it as if it was yesterday.”*


Consequently, the way the first interview and subsequent interventions were structured depended on how the participants intentionally engaged with trauma during initial therapeutic encounters and the varied ways they re-called experienced victimization at different ages. This varied as a visceral embodied experience when recalling trauma at the age of nine years for Violet, who shared:


*“It feels as if I’m trapped in my body; it feels stiff. I feel like something is pushing down over the lower part of my tummy.”*


For Joan, accessing an auditory memory of the perpetrator (her uncle), who molested her from the age of eight years, and only disclosed the trauma for the first time during her first session, shared:


*“I can clearly hear his voice; he is standing in the passage, I can clearly hear his footsteps coming down the passage before he comes into my room. I know what is coming; he wants to touch me.”*


Debra, recalling how her body felt 42 years after being raped at the age of 19 years:


*“I felt like I was decaying and infected.”*


Within the ITIC therapeutic framework, relational trust was not presumed but gradually co-constructed through the consistency, sensitivity and ethical responsiveness of the therapeutic relationship. For the participants, non-disclosure and emotional restraint were often reinforced and, consequently, the therapeutic space served as a critical site of safety and stabilization. Within this environment, participants were able to gradually relearn relational and emotional safety, particularly around authenticity and self-expression that had previously been discouraged, with the aim of making these capacities transferable to other aspects of their lives and relationships. Both reflective questioning and follow-up sessions highlighted the importance of the positioning of the primary researcher as therapist, in not facilitating participants’ trauma-related tendencies of learned helplessness and co-dependency [40]. Feedback from the participants further illustrated the importance of facilitating the therapeutic process in this way. Debra shared the following:


*“I’m not sure what you do, but it works; I feel much better. I know that I was safe and could share what had happened to me. I didn’t try to control everything so much.”*


Similarly, following her disclosure of repeated abuse, Zara testified against her maternal grandfather during legal procedures; however, he was ultimately acquitted. The Muslim community she belongs to knew what happened to her but never openly spoke about it. She shared the following, after the therapist guided her to deeper authenticity through non-directive probing:


*“What do I deserve? No one has ever asked me these questions as you do. I just always felt like this poster child in my community but no one really spoke about it, though everyone knew what had happened to me.”*


Aisha, a Sunni Muslim, growing up in a community where sexual trauma was stigmatized, shared the following:


*“I am enough; it was difficult to be so vulnerable and accept myself. But this process has shown me it is ok to be honest.”*


Charlotte, who struggled with severe anger towards her father who abused her, and who attended eight years of dialectic behavioural therapy sessions, the embodied, affective integrative ITIC process assisted her in expressing the following:


*“Now I feel fine; I have never experienced anything like this; it is difficult to be honest and feel the effect in my body but I am so relieved.”*


It was evident that the participants had ways of coping and dissociating from the long-term effects of what they have experienced since childhood, ranging from isolation, deep emotional dissociation and distress to religious coping-styles [33]. For example, Sarah, who was sexually abused since the age of five in her family home by a visitor and repeatedly abused until the age of 18 and raped by a gang leader at 17 years, shared the following:


*“I will just have a poker face; inside it will be turmoil but you will never see it. I want to be on my own but I feel lonely at the same time”.*


These complex traumatic lived experiences of CST necessitated the co-development of grounding practices as part of the ITIC process.

### 3.2. Grounding Practices

Establishing a foundation of safety and agency, both within the body and in the therapeutic environment, emerged as a central theme in the implementation of the ITIC methodology. This principle was enacted through several interrelated practices founded on the intentional cultivation of ongoing therapeutic relational trust described above. The intervention techniques described in Appendix A include “Hold space—Emotion”, “Embodied awareness”, “Acknowledge”, “Reframe” and “Allow emotion”, reflecting a non-directive trauma-informed approach that prioritizes emotional safety and ample space and time for trauma awareness and processing. Rather than guiding the process through prescriptive techniques, these responses created open-ended opportunities for participants to gradually become aware of, express and integrate trauma at their own pace in a manner grounded in relational presence and embodied responsiveness. The setting of clear, consistent and supportive interpersonal boundaries that involved collaboratively defining the structure, pace and expectations of each session helped participants to develop a sense of predictability and control. The role of integrative silence in this process was paramount [34]. Boundaries were not imposed, but co-negotiated, offering participants the opportunity to explore safe relational dynamics, often for the first time, within a therapeutic context. This process also modelled boundary-setting skills that could be transferred to other areas of their lives. For example, the therapist reassured Mandy, who had experienced repeated abuse by her stepfather that led to a pregnancy, in the following way:


*“So, now is the time for you to start learning what it means to take care of yourself. Mom abandoned and rejected you and you went through the worst of traumas. What happens to your mind is it protects you by creating a barrier, like a wall, between you and what happened to you, between who you truly are as a person and what happened to your body. The barrier is created for your survival and then you start living in survival mode and don’t care about yourself. In this therapy, we address and include who you are as a whole person. You don’t have to revisit every memory to recover but we include your body and emotions in the process, for example, you feel frustrated as if you always give too much of yourself. This often happens when you had such a history; you keep on giving to your own detriment and then that can leave you feeling like an island, as you shared. So, your boundaries need to heal up as well. I create space and time for you to connect with what is necessary and you can learn to trust yourself in the process.”*


In addition, the acknowledgement of structural violence, community-specific histories and the lived realities of intergenerational trauma exacerbated by socio-economic hardship was essential to contextualizing individual trauma within broader systems of oppression. This is illustrated by Mandy’s experiences of sexual abuse, as she was left at home alone for extended periods of time by her working mother, which made her particularly vulnerable to being repeatedly tied up and raped by her stepfather. Validating these realities allowed for a de-pathologizing of trauma responses and opened up space for critical conversations around resilience, survival and resistance, as shared with Mandy. The co-creation of grounding practices tailored to individual needs allowed participants to actively shape their own healing processes—for Mandy, it was spending time in her garden and caring for pets in the community. These practices included sensory awareness and movement, developed collaboratively based on what felt safe and effective for each participant. By centring the body as a site of wisdom and healing, these practices supported emotional regulation and reconnected participants to a sense of embodied awareness and agency.

### 3.3. Embodied Awareness

Reconnecting the body with personal narrative, thereby promoting agency and ownership of lived experiences, emerged as a central therapeutic process within the ITIC framework. Somatic practices that helped participants attune to their bodily sensations included breath, posture, tension, discomfort and allowing pain. Trauma narratives were explored at each participant’s pace, allowing for the gradual integration of dissociative and fragmented experiences. This process was further deepened by recognizing intergenerational and culturally embedded trauma patterns, which provided a broader context for individual healing and supported a decolonial approach to trauma care. Study participants were encouraged to voice their embodied experiences, creating space for authentic self-expression. Amber shared the following:


*“I feel very heavy, like lead. My head is not ok. I’m always sad … I don’t say anything. My heart pains and just implodes.”*


Debra, who was sexually assaulted at knifepoint, shared the following:


*“The sadness is everywhere in my body.”*


Aisha, during the initial stages of the ITIC process, when disclosing the sexual grooming she experienced as a six-year-old at the hands of a group of boys in her community, shared how her body felt during recalling and disclosing what had happened to her:


*“I’m feeling overwhelmed. It sits in my chest.”*


For Violet, disclosing the abuse from her uncles and a friend of her father’s led to immediate reactivity in her body, which she experiences as the following:


*“I feel vibration, discomfort in my whole body and pressure in my reproductive parts”.*


Joan, who had never disclosed the trauma she endured from her uncle and had suppressed the trauma until the first session, shared the following:

*“I can feel my heart beating now; I feel as if I was dead.”* (She starts crying.)

Amber, who suffers from diagnosed anxiety, depression and many physical ailments including asthma, described the way she felt after one experience of remembered trauma integration:


*“I am at peace. My chest feels much better. The birds are singing; everything in my head is open, white and clear”.*


These unique participant-specific lived, embodied experiences influenced the way trauma integration led to adaptive psychosocial regulation.

### 3.4. Adaptive Regulation

Equipping participants with tools for sustainable self-regulation and meaningful social connection was a central objective within the ITIC framework. This was supported through body-centred practices which enhanced participants’ capacity to notice and respond to internal cues. Developing emotional literacy and the ability to express emotions safely further strengthened self-regulation. Additionally, the process of rebuilding safe relationships and re-establishing interpersonal boundaries enabled participants to apply these skills in their everyday social environments, strengthening long-term integration and relational resilience.

Emotional awareness was explored after establishing a safe and trusting therapeutic space; sadness, fear and anger were acknowledged as valid expressions of embodied trauma and implicit memory recall. For example, the researcher further explored Sarah’s experience of “inner turmoil”, enquiring, “Okay, tell me about the inner turmoil. What happens? What does it feel like?”, to which Sarah replied, “Loneliness, you know”. The researcher reflects this response back: “and the closest emotion to loneliness is what? Sadness, fear, anger or something else?”, to which Sarah replied, “It is sadness. It sits in my heart, so painful”, which illustrates the integration process.

### 3.5. Integration

From a critical feminist perspective, the researcher examined the context-specific societal factors that shaped the participants’ trauma, while holding continuous space for embodied awareness during the integration process. An emphasis on the continuity of recovery, conceptualised as an ongoing, relational process rather than a fixed or linear endpoint, was one of the main principles guiding the ITIC approach. Central to this perspective was the integration of embodied awareness, which played a critical role in addressing the dissociative patterns commonly associated with long-term sexual trauma suppression. Lydia’s experience of disclosing and integrating the trauma of racial, physical and religious abuse she endured as the child of a domestic worker in the house of a white, Afrikaner family, were reflected as follows:


*“The mountain is my life; it went into the blue water and disappeared. My sadness is gone; there is so much peace in my body now.”*


Joan, who relied on her faith and prayer, described her experience of embodied integration as follows:


*“My body feels much calmer now, much better.”*


Ava, who experienced rape and public gangrape and recovered from a light stroke between sessions six and seven, said:


*“My body feels relieved; I am at peace.”*


Violet, who shared the experiences of sexual grooming and abuse since the age of nine years, at the end of the first session shared the following:


*“You have no idea how I felt after that first session; it felt as if I could breathe again. A dark shadow was hanging over me all the time.”*


Embodied awareness allowed participants to identify and process sensations linked to pain, fear and disconnection, with a renewed sense of safety and agency within their own bodies. This deeper somatic engagement supported participants in moving beyond survival-based coping mechanisms, enabling them to reflect meaningfully on their personal growth, acts of resistance and the reclamation of identity. For Sarah, recovery also unfolded through her academic journey, which became a site of liberation rooted in social justice and faith-based inquiry. While all participants expressed some hesitation in engaging in community-based activism due to the social visibility it requires, their reflections highlight the broader social dimensions of healing [41]. The support and activities offered by the Bonteheuwel Walking Ladies, a community-based activist group, helped to address this hesitation and encourage sustained social engagement. However, participants from other areas of the Cape Flats, such as Heideveld and Lavender Hill, did not have access to similar support structures, due in part to transport limitations, financial constraints and the impact of community-specific violence. This illustrates the importance of the establishment of ongoing community support structures.

The co-construction of relational safety, the centrality of grounding and somatic practices, phase-based embodied awareness and ongoing adaptive regulation are consistent with and extend contemporary evidence on trauma care for survivors of sexual violence. Systematic reviews indicate that psychosocial interventions delivered flexibly within trusted therapeutic relationships reduce PTSD and depressive symptoms and work best when tailored to clients’ contexts and pacing [42]. The qualitative and clinical literatures have indicated that body-centred practices and processes can access implicit memory and affect regulation beyond purely narrative exposure [34]. A decolonial and feminist lens that foregrounds survivors’ situated, embodied knowledge is especially pertinent in marginalized settings and supports contextualizing individual trauma within structural violence and intergenerational patterns [43]. Regional reviews of sexual and gender-based violence (SGBV) in sub-Saharan Africa further confirm that survivors face complex health sequelae and that service gaps (including limited community supports and resource constraints) shape recovery trajectories and the feasibility of sustained community mobilization [43,44].

## 4. Discussion

The ongoing presence of structural violence and trauma within communities was made evident by the exacerbated physical vulnerability of the women included in this study. When they were children, they had to find ways to navigate blurred embodied boundaries, sharing family homes and community spaces with sexual perpetrators where sexual victimization was ignored, shamed and silenced. This had context-specific implications for the way trauma-informed care was made available and the efficacy of interventions evaluated. Progress towards emotional integration and self-care was not uniform—some participants experienced a gradual shift towards peace and empowerment, while others encountered only fleeting moments of relief. Healing from CST is complex, non-linear and may be uneven, with long-term outcomes remaining uncertain, particularly for marginalized communities like the Cape Flats, where socio-economic hardship, gendered violence and systemic inequality continue to affect well-being.

As community violence and endemic sexual violence are at the heart of feminist concerns in South Africa [31,41], the main research objective was to provide specialist trauma support for the study participants. As Zara (age 21 years), the youngest participant, noted, while making an important observation reflecting the intergenerational legacy of apartheid:


*“There is a backlog due to apartheid. People are carrying severe trauma in community. There is really no change since democracy”.*


The societal scars of deep-seated trauma speak to the realities of both current day South Africa and contexts around the world, as exemplified in the #Blacklivesmatter and #Metoo movements. The authors argue that producing feminist, situated knowledge transcends disciplinary boundaries, particularly when considering community-specific, integrative health care. The importance of deconstructing the therapeutic space in this way allows reflexivity and critique at every moment that interventions are applied [30].

Given our findings, the co-development of therapeutic pathways in context-sensitive trauma-informed care illustrates a non-linear yet progressive participant-specific journey. The colour-coded therapeutic care pathway sequencing included in Appendix A offered detailed, time-coded observations from recorded sessions, clinically verified by the first author, a psychologist with extensive experience in complex trauma intervention assessments. Using critical analysis methods, deduction, reasoning, interpretation and analysis, the authors, through critical discussions and analysis, inferred that participants’ improvements were supported by the ITIC intervention.

This is based on ITIC principles including relational safety, embodied awareness and participant-led trauma integration pacing. The study participants’ experiences collectively demonstrate how trauma integration and recovery unfold through various stages, often involving initial disclosures, somatic processing, emotional regulation and the integration of past experiences into a coherent narrative. The format described in Appendix A can be considered to provide guidance for clinical trauma-informed assessments and intervention planning. In addition, we suggest practitioners consider the following six therapeutic principles as part of community-specific care.

### 4.1. Safe, Trauma-Informed Awareness

Initial therapeutic engagement for many participants began with spontaneous disclosures of the lived experiences of sexual violence. For example, Sarah, Charlotte, Joan, Debra and Mandy initiated their processes with limited therapeutic prompting, met with a supportive therapeutic stance, described as “Holding Space” in Appendix A. These moments established relational safety and enabled early exploration of affective, embodied dissociation and emotional distress. The initial sessions revealed strong emotional themes (e.g., fear, anxiety, objectification, sadness) and somatic symptoms including chest, heart and stomach pain for Sarah, Ava and Evelyn and intense fear with bodily trauma recall for Mia. This embodied processing and somatic awareness indicate main therapeutic transition moments, occurring as participants moved from cognitive processing to embodied awareness. Sarah, Mia, Lydia, Ava, Evelyn, Aisha, Mandy and Debra all showed increasing somatic–emotional integration. This included full-body anger, visceral memory recall and localized sensations (e.g., sadness in the heart), reflecting trauma integration through the body—a central goal of the co-developed care pathway therapeutic methodology.

### 4.2. Empowerment Through Personal Agency

Central to feminist trauma care is the emphasis on autonomy and self-determination. The therapeutic responses (e.g., “Authenticity Reframe”, “Reflect Back”, “Hold Space–Embodied”) affirm a strengths-based, co-constructed process that honours both the pain of trauma awareness and agency for each participant. The return to “Integration” in multiple sessions shows that the recovery process was characterized by trauma-informed shifts emerging through repeated, embodied exploration. Sarah’s process exemplifies how the ITIC methodology allowed for personalized, relationally anchored trauma care, moving from fragmented, shame-based narratives toward embodied empowerment and an integrated identity, which included her religious beliefs. The structured yet flexible application of thematic phases and therapeutic responses are indicative of the efficacy of contextually grounded, feminist-informed approaches for adult victim-survivors navigating the layered impacts of trauma, particularly from contexts marked by spatial precarity and intergenerational violence.

### 4.3. Collaborative, Non-Hierarchical Therapeutic Relationship

Within a feminist-centred trauma care model, the therapist acts as a collaborative partner. Victim-survivors are encouraged to share their experiences without fear of judgement or pathologization, thereby strengthening a sense of empowerment and control over their recovery journey. Ava benefited from a therapeutic space that enabled grounding and safe containment of emotional authenticity. For Charlotte, trust was essential due to her ongoing struggles with mistrust, and Mandy was supported in reinforcing her agency and self-trust, particularly through the therapist’s acceptance of her choice not to revisit certain trauma memories. Evelyn began exploring interpersonal boundaries, marking an important developmental shift toward relational agency.

### 4.4. Contextualizing Trauma Within Systems of Oppression

The study illustrates that CST often intersects with other forms of oppression, such as gender-based violence, racism and economic inequality. An integrative feminist approach contextualizes trauma not only as an individual experience but also as a product of social systems. Consequently, the therapeutic process engaged participants in confronting complex and deeply personal themes. Sarah explored feelings of being an outsider (black sheep) and experiences of darkness and suicidal ideation, while Mia processed sexual objectification, anger and fear. Ava addressed traumatic experiences involving incest, gang rape and emotional numbness. Charlotte worked through patterns of co-dependence, mistrust and unresolved anger, and Joan navigated fear, dissociation and shame. Their coping strategies was respected while supporting deeper emotional integration. For instance, Joan’s and Lydia’s use of religious dissociation and guilt were held without judgement, enabling further exploration. Aisha initially relied on spiritual and cognitive defences, which gradually gave way to more authentic embodied–affective trauma integration.

### 4.5. Validation and Self-Advocacy

Part of emphasizing personal agency is reinforcing the victim-survivor’s ability to set boundaries, express their needs and seek recovery on their terms, whether through community support or personal transformation. The therapeutic process supported participants in reclaiming internal safety, agency and ownership of their narratives. Ava’s later sessions showed increased vitality and assertiveness and Zara’s act of naming her perpetrator signified personal acknowledgement and ownership of her story. Alongside this, participants showed progressive shifts from emotional dysregulation to greater coherence and integration. Sarah’s experience evolved toward peace and empowerment, while Mia developed emotional integration and self-care. Joan accessed moments of calm and integration, and Lydia moved from silence and shame to embodied emotional clarity. Evelyn’s reflections on boundaries and affective expression marked integration, while Aisha’s journey concluded with peace and wholeness. Amber, Debra and Violet each experienced moments of peace, indicating trauma integration.

### 4.6. Practice and Policy Implications

As religious organisations and community activist support groups often attempt to provide sufficient psychosocial support in the context of very limited access to specialist mental health care, the authors propose improving collaboration. Advance shared decision-making to ensure that care is culturally sensitive and responsive to the unique needs of the community. Strategies include: (i) co-designing care models by considering the study findings, involving community members in the design and implementation of trauma care programmes, ensuring that their voices, experiences and values are central to the process; (ii) building capacity by providing training and resources to local support groups, enhancing their ability to offer trauma-informed care and integrate mental health support within the community context; (iii) strengthening community trust by establishing long-term, consistent relationships with community activists and leaders; (iv) offering culturally tailored interventions by adapting trauma care strategies to the community’s cultural, spiritual and social context, recognizing the importance of local traditions, values and healing practices [33]; and (v) creating safe, inclusive spaces where trauma survivors feel respected, heard and empowered to engage in their healing process.

The results highlight the need for trauma-informed, culturally grounded and survivor-centred care pathways, supporting mental health practice and public health policy that not only include biomedical models of care but include the embodied, relational and structural dimensions of trauma integration and recovery. By integrating feminist and decolonial approaches, policies can more effectively reduce barriers to care as part of community health, promote equity and ensure that the voices of marginalized survivors inform service design and implementation.

## 5. Conclusions

The implementation and assessing of the research methodology designed to facilitate embodied, community-accessible, trauma-informed care within an integrative mental health framework was outlined. By prioritizing feminist approaches, this study offers a nuanced methodology tailored to adult survivors of sexual trauma within intergenerationally marginalized communities in Cape Town. The approach indicates the necessity of integrating systematic care across diverse socio-ecological contexts and highlights the importance of participant agency and contextually relevant mental health care for women affected by colonial and gender-based oppression. It further emphasizes the necessity of accountability in research through alternative epistemologies [9] and the integration of diverse perspectives was pivotal in exploring alternative discourses in community-based trauma care [18,30,36]. Hence, the research methodology explored the critical need to expand the vistas of trauma, social justice, historically situated binaries of socially marginalized identities, structural power and related oppression in the South African context.

### Study Limitations and Recommendations

The current findings were based on an in-depth qualitative analysis of 13 participants’ lived experiences in engaging this therapeutic methodological assessment. Therefore, including participants from diverse regions and populations for further research and clinical practice is important. These findings may be considered as a pilot study from which care and training models can be developed, based on lived experiences, intervention timelines, lifespan comparisons, age differences and specific community needs.

In this pilot study, the interventions were co-developed by engaging the study participants, though additional cultural insiders were not formally involved. Capacity-building through training or resourcing of local support groups was not implemented, and future efforts could include structured workshops, mentorship and resource provision to strengthen community support. The intervention was culturally and socially adapted by incorporating participant narratives and local idioms, with further development potentially benefiting from consultation with spiritual leaders, inclusion of culturally specific coping practices and iterative community feedback to enhance relevance. On the Cape Flats, ongoing support for survivors could extend beyond formal activist groups to provide logistical support to existing survivor support networks, including Rape Crisis and the government-funded Thuthuzela Care Centers (TCCs), which are one-stop facilities designed to provide comprehensive care and support to survivors of sexual violence. Practical efforts to broaden support structures might involve establishing partnerships with these community actors, providing basic trauma-informed care training to church leaders and traditional healers, creating peer-led survivor groups within neighborhoods and facilitating safe community spaces where survivors can share experiences and access resources. Leveraging mobile or digital platforms could also help connect survivors in low-resourced areas with broader support networks while accommodating safety and accessibility concerns.

## Figures and Tables

**Figure 1 healthcare-13-02601-f001:**
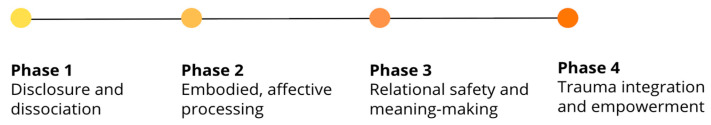
Phased timeline of trauma awareness and integration.

**Figure 2 healthcare-13-02601-f002:**
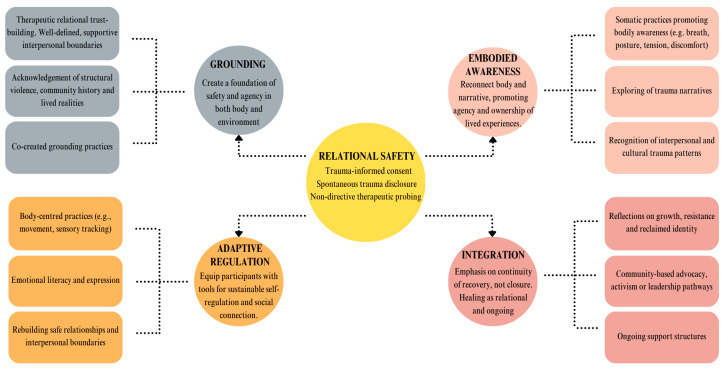
Therapeutic care pathway co-development: colour-coded themes and subthemes identified from reflexive, inductive analysis.

**Table 1 healthcare-13-02601-t001:** Rationale for semi-structured and unstructured interview format with clinical observations.

Technique	Description	Rationale for Use in Complex Trauma Integration
Semi-Structured Interviews	Guided by clinical assessment themes (trauma history, psychosocial functioning, coping strategies) while allowing flexibility in question order and pacing.	Provided a safe, predictable structure while allowing space for participants to share at their own pace; supported rapport-building and careful timing of sensitive questions to avoid re-traumatization.
Unstructured Interviews	Open-ended conversations allowing participants to direct the flow, share memories spontaneously and introduce personal significant themes.	Enabled participants’ agency and “coming to voice”, allowing disclosure of suppressed trauma experiences and the co-construction of meaning in a non-pathologizing way.
Clinical Observations	Ongoing attention to non-verbal cues (body posture, tension, facial expression), tone of voice, silences and emotional shifts during sessions.	Informed therapeutic pacing and intervention timing; supported embodied awareness, recognized dissociative states and ensured emotional safety throughout the process.

**Table 2 healthcare-13-02601-t002:** Observation techniques in clinical assessment of embodied trauma (sexual abuse context).

Technique	What Is Observed	Clinical Purpose
Body Posture/Movement	Collapsed posture, guarded stance, fidgeting, rigidity, freezing or avoidance of eye contact.	Identifies somatic manifestations of fear, shame or hypervigilance, and signals readiness or overwhelm.
Facial Expressions/Micro-Expressions	Flinching, tearfulness, blank stare, sudden muscle tension, or fleeting expressions of fear/disgust/anger/sadness.	Detects unconscious emotional responses and moments when trauma memory may surface.
Tone/Pace/Volume of Voice	Shifts from monotone to rapid speech, whispering or silence.	Provides insight into emotional activation, dissociation or hyperarousal states.
Breathing Patterns	Shallow, irregular, held breath; sighs or deep exhalations.	Indicates autonomic nervous system regulation and helps time grounding interventions.
Silence/Pauses	Prolonged silence, hesitations or sudden speech cessation.	Creates therapeutic space for memory processing; signals emotional intensity or dissociation.
Physiological Cues	Sweating, trembling, flushing, restlessness or visible tension in specific body regions.	Reveals somatic activation linked to traumatic recall and informs pacing of intervention.
Eye Gaze/Orientation	Looking away, fixed gaze, darting eyes or closing eyes.	Suggests avoidance, safety-seeking or accessing internal imagery and memory recall.
Somatic Descriptions by Participant	Reports of pain, heaviness, numbness, pressure or “trapped” sensations.	Validates embodied memory and guides somatic-focused interventions.
Affect/Emotional Shifts	Sudden anger, withdrawal, tearfulness or laughter.	Helps identify emotional triggers and opportunities for regulation and integration.
Relational Cues	Dependency, over-compliance or difficulty trusting therapist.	Informs relational attunement and co-construction of a safe therapeutic alliance.

## Data Availability

The data presented in this study are available upon request from the corresponding author due to the sensitive nature of the research. (While participants provided written informed consent for anonymized research data, they did not give written consent for the original data set to be shared publicly.)

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
