# Peer review of "Healing Bodies, Healing Communities: A Community-Based Qualitative Study of Adult Survivors of Childhood Sexual Trauma in South Africa"

_healthcare, 2025, doi:10.3390/healthcare13202601_

Round 1

Reviewer 1 Report

Comments and Suggestions for Authors

Thank you for allowing me to review the article titled: “Healing Bodies, Healing Communities: Co-developing Therapeutic Care Pathways for Adult Survivors of Childhood Sexual Trauma in South Africa”

Title

It is recommended to include the type of study in the title, for example: “A Participatory, Community-Based Qualitative Study.”.

Abstract

Present the methodology more directly, avoiding duplication.

Describe the findings and results in a more concrete way.

Briefly explain how the study’s findings could be applied beyond the African context.

Introduction

Highlight more explicitly the justification related to the lack of appropriate care models for survivors.

Clearly state the aim of the study and emphasize the importance of co-creating adapted therapeutic models.

Make evident that the study was collaborative and participatory, underlining the importance of actively involving survivors in the process.

Methodology

Clarify the participatory model, detailing how participants contributed to the co-creation of the care model.

Provide more information on participants’ sociodemographic and health characteristics, as well as the selection process (type of sampling according to qualitative research standards).

Give more detail on data collection techniques (type of interviews, type of observation conducted).

Explain how feminist and decolonial approaches informed the research, ensuring reflexivity.

Describe the inductive analysis process, validation of results, and how qualitative rigor was maintained (credibility, transferability, reliability).

Results

Include participant testimonies or examples showing how key results were applied in practice.

Strengthen the connection of results to theoretical frameworks such as epistemic injustice, feminist, and decolonial approaches.

Link therapeutic interventions directly with the observed improvements.

Simplify the presentation of Table 1. Consider moving the full version to the supplementary material.

Discussion

Reinforce the findings by comparing them with previous studies.

Explore how the results may influence public health policies.

Discuss how the findings can be applied in mental health, community health, and feminist approaches.

Conclusion

Emphasize the need for culturally and community-congruent care models, highlighting their relevance for future research design.

Mention the potential application of these models in other regions and with different populations.

Reviewer 2 Report

Comments and Suggestions for Authors

COMMENT 1: Although the value of critical self-reflection is highlighted at various points in the paper, such self-reflection is, however, largely described in relation to “finding participants at the participant’s level of disequilibrium”. As such, relatively little attention is paid to critical reflection in relation to “finding the researcher at the researcher’s level of disequilibrium”.  In fact, the ‘person’ behind the words written in this paper remains largely ‘masked’ with the exception of one or two cursory comments (e.g.,  the researcher “ remained attentive to her presence as a white, Afrikaans therapist”, lines: 174-175), which raises a number of questions in my mind:

  • Who is this researcher in the context of the research?
    • Was the researcher in any sense a cultural/social 'insider' in relation to participants?
    • What strengths and vulnerabilities did the researcher bring to the process of research engagement?
    • What (if any) was the emotional impact on the researcher of engaging with this research?
    • To what extent did the researcher engage in critical self-reflection in relation to (a) the possible impact of conducting research on a ‘sensitive’ topic, and (b) self-care strategies that could have been employed to prevent or to mitigate the effects of research-induced distress on the researcher.

To the extent that the researcher’s experience of research engagement constitutes important data (in its own right), it can only enhance the paper (and provide important insights for other researchers) if the authors were to “raise their masks” just a little.

COMMENT 2 (lines: 586-588): According to the authors, intervention strategies should ideally include:

  • Involving community members in the design of trauma care programs. Please clarify which community members were involved in the design of this intervention. Did this involve only participants and/or community service providers, or were additional cultural-insiders involved in the design phase? Please clarify
  • Building capacity by providing training and resources to local support groups. Was this done? And if not, what practical recommendations can the authors make in this regard.
  • Adapting interventions to the communities cultural, spiritual, and social context. What specific adaptations were made in this regard, and what (if any) additional adaptation strategies may be indicated in the further development of the intervention model. Please clarify.

[The authors acknowledge that the intervention strategy reported in this study may be regarded as a pilot study (line 615-617) – with efforts to address the above issues (inter alia), as well as practical recommendations for doing so, being likely to provide a clearer indication of the way forward].

COMMENT 3: The authors point out (477-483) that not all participants had access to community-based activist groups for ongoing support (an occurrence that is not unusual in low resourced settings). Are there other potential sources of ongoing support (e.g., churches, traditional healers, survivor support groups, etc.) that could be harnessed to address the gap? And what practical efforts could be made to broaden ongoing support structures?

 COMMENT 4 (lines 575-580): It would appear that participants’ progress towards emotional integration and self-care was not uniform – with some participants’ experiences shifting progressively towards peace and empowerment, and with some participants experiencing only moments of peace. While such variations in outcomes are not surprising, I am wondering if the authors have any suggestions regarding the dynamics of these differential outcomes (just an idea that the authors may wish to consider).

Round 2

Reviewer 1 Report

Comments and Suggestions for Authors

Thank you for allowing me to review the article titled: Healing Bodies, Healing Communities: A Community-Based Qualitative Study on Adult Survivors of Childhood Sexual Trauma in South Africa.

Reviewer 2 Report

Comments and Suggestions for Authors

No comment